# Magnetic Self-Healing Composites: Synthesis and Applications

**DOI:** 10.3390/molecules27123796

**Published:** 2022-06-13

**Authors:** Kenneth Cerdan, Carlos Moya, Peter Van Puyvelde, Gilles Bruylants, Joost Brancart

**Affiliations:** 1Department of Chemical Engineering, Soft Matter, Rheology and Technology (SMaRT), KU Leuven, Celestijnenlaan 200J, 3001 Heverlee, Belgium; kenneth.cerdangomez@kuleuven.be (K.C.); peter.vanpuyvelde@kuleuven.be (P.V.P.); 2Engineering of Molecular NanoSystems, Ecole Polytechnique de Bruxelles, Université Libre de Bruxelles (ULB), Avenue F. D. Roosevelt 50, CP165/64, 1050 Brussels, Belgium; gilles.bruylants@ulb.be; 3Physical Chemistry and Polymer Science, Department of Materials and Chemistry, Vrije Universiteit Brussel, Pleinlaan 2, 1050 Brussels, Belgium; joost.brancart@vub.be

**Keywords:** magnetic self-healing composites, magnetic fillers, magnetic (nano)particles, synthesis, processing, manufacturing, health, actuators, stretchable electronic, slippery surfaces

## Abstract

Magnetic composites and self-healing materials have been drawing much attention in their respective fields of application. Magnetic fillers enable changes in the material properties of objects, in the shapes and structures of objects, and ultimately in the motion and actuation of objects in response to the application of an external field. Self-healing materials possess the ability to repair incurred damage and consequently recover the functional properties during healing. The combination of these two unique features results in important advances in both fields. First, the self-healing ability enables the recovery of the magnetic properties of magnetic composites and structures to extend their service lifetimes in applications such as robotics and biomedicine. Second, magnetic (nano)particles offer many opportunities to improve the healing performance of the resulting self-healing magnetic composites. Magnetic fillers are used for the remote activation of thermal healing through inductive heating and for the closure of large damage by applying an alternating or constant external magnetic field, respectively. Furthermore, hard magnetic particles can be used to permanently magnetize self-healing composites to autonomously re-join severed parts. This paper reviews the synthesis, processing and manufacturing of magnetic self-healing composites for applications in health, robotic actuation, flexible electronics, and many more.

## 1. Introduction

Magnetic (nano)particles and derived magnetic composites show changes in their properties (e.g., magnetorheology), structures or positions/motion (electromagnetic transduction) in response to the application of an external magnetic field or as a consequence of inherent magnetic properties [1,2,3,4]. This has triggered the attention of many researchers during recent decades for exploitation for a wide range of different applications. The response of the magnetic composite depends on the type and structure of the magnetic filler used, the properties of the matrix material and the interaction between the magnetic filler and the inert matrix. Hard magnetic (nano)particles can be permanently magnetized by an external magnetic field, whereas soft magnetic (nano)particles lose their magnetization upon removal of the external magnetic field. Magnetic polymer (nano)composites are created using a variety of magnetic particle fillers with different magnetic properties and processed using different physical and chemical techniques, leading to different nanocomposite morphologies and different levels of complexity in their responses to applied magnetic fields.

Self-healing materials have been drawing much research interest in the past decades. They exhibit the ability to recover their functional properties by healing incurred damage, thus extending their lifetimes and those of the objects and structures made from these self-healing materials [5]. Two fundamentally different healing mechanisms can be distinguished. Extrinsic self-healing materials rely on a healing agent that is stored in hollow fibres [6], capsules [7] or microvascular networks [8], mimicking the cardiovascular systems seen in nature. When these reservoirs break due to mechanical damage, the healing agent leaches out of the reservoirs, filling the damage volume. Upon solidification, usually following a chemical reaction, the mechanical properties of the material are recovered. Extrinsic self-healing mechanisms are almost exclusively used for thermosets and related composite materials. Alternatively, the healing ability of intrinsic self-healing materials is inherent to the chemical structures of the materials. Intrinsic self-healing polymers have reversible covalent bonds (e.g., reversible cycloaddition [9], metathesis or exchange reactions [10,11]) or noncovalent interactions (such as hydrogen bonds [12] or ionic interactions [13]) built into their polymer network structures. These reversible bonds and interactions are broken preferentially and in a reversible fashion upon mechanical overloading. After the removal of the mechanical load, these bonds can be reformed either autonomously or upon the application of an adequate stimulus, such as heat or light irradiation. Intrinsic healing systems are used for thermosets, elastomers and derived composites. In thermosets, usually, a thermal trigger is necessary to create the chain segmental mobility required for the effective healing of the incurred damage, whereas certain healing chemistries enable the autonomous damage healing in elastomers. In this regard, the formulation of self-healing magnetic composites has the potential to solve major issues in the respective fields of magnetic materials and related applications and in the developing field of self-healing materials.

This mini-review focuses on this intriguing class of smart, functional materials that can respond to many types of stimuli. The ability to repair damage and thus extend the service lifetimes of materials, systems and structures is expected to find application in many different fields. As self-healing materials have significantly matured in the last decades, the focus has shifted towards the recovery of a wider variety of combinations of functional properties that may be lost after damage. Magnetic (nano)particles add unique capabilities to the derived material composites, which find applications as, e.g., sensors or electromagnetic actuation mechanisms in electronic devices, transport, environmental, biomedical and many other fields. Recent literature reports on self-healing magnetic composites were reviewed in view of these and other applications. On the other hand, magnetic properties can also be employed to activate the self-healing ability or to serve as an aiding mechanism to improve the self-healing performance. Terryn et al. critically assessed a broad range of healing chemistries and mechanisms and related self-healing elastomers for soft robotic applications [14]. They identified that magnetic elastomeric composites have the potential to fabricate self-healing magnetic actuators that are able to recover their magnetic behaviour and actuation performance. Furthermore, a more important discussion was held regarding the potential of magnetic particles to aid the healing mechanism through the potential to close large damage sizes.

Magnetic polymer (nano)composites are created using a variety of magnetic particle fillers with different magnetic properties and processed using different physical and chemical techniques, leading to different nanocomposite morphologies and different levels of complexity of their responses to applied magnetic fields. The synthesis and production techniques are reviewed, particularly in view of the fabrication of self-healing magnetic composites and how the choices of polymer matrices, magnetic particles and processing methods and conditions affect the final composite properties.

## 2. Magnetic (Nano)Particles

The magnetic properties of self-healing magnetic composites stem from the magnetic (nano)particles used to create the composites and how they are dispersed in the self-healing polymer matrix. The properties of the magnetic (nano)particles depend on their chemistry but also on the synthesis method used, as this will determine the particle shape, size (distribution) and crystal structure.

### 2.1. Synthesis Methods

In the last two decades, several synthetic routes have been developed to prepare magnetic (nano)particles with good control over the particle size, polydispersity, shape, crystallinity and magnetic properties [1,2]. Top-down and bottom-up approaches are the two main strategies for preparing magnetic (nano)particles. The former starts from macroscopic structures whose size is reduced in the nanoscale range by physical methods such as, among others, laser evaporation [15], ball milling [16,17] and wire explosion [18]. The latter, generally based on biological or chemical routes, have been much more explored due to the lower cost of production, greater reaction yields and better spatial control over the system’s characteristics. Among the chemical routes, coprecipitation and thermal decomposition were the first to be explored and became the most popular due to their improved performance compared to other strategies (see Figure 1) [2,19,20]. As the most conventional route, coprecipitation is based on mixing metal salts in an alkaline medium without surfactants [19,20]. Temperature, precursor concentrations, reaction time and stirring are the key parameters that determine the final particle characteristics. The advantages of the coprecipitation route are its simplicity, low price and great efficiency, allowing for the production of magnetic (nano)particles in large amounts. Nevertheless, the systems synthesized by this method usually show poor control over the particle morphology, a broad size distribution, and the tendency to form particle agglomerates that hinder the control over their dispersion into the polymeric matrix. This methodology has been widely explored for the synthesis of ferrite magnetic (nano)particles in sizes ranging from a few to hundreds of nanometres. In particular, magnetic iron oxide nanoparticles (NPs) composed of magnetite (Fe_3_O_4_) or maghemite (γ-Fe_2_O_3_) have been among the most commonly studied because of their ease of production, low toxicity and good magnetic properties [21,22].

Thermal decomposition employs, on the other hand, high temperatures to decompose organometallic precursors in the presence of organic surfactants, e.g., fatty acids, amines and alcohols, using organic solvents with high boiling points [23]. Generally, this route allows for the production of magnetic (nano)particles with an excellent control over the structural properties such as a narrow size distribution, a well-defined particle morphology and a good crystallinity. However, their main bottlenecks are the elevated price per synthesis, the high energy consumption and the difficult control over the complex reaction profiles that involve many reagents. In a similar way to coprecipitation, this synthetic route has proven to be ideal for the preparation of magnetic iron oxide (nano)particles in a wide range of sizes [24,25,26].

### 2.2. Magnetic Properties

The magnetic properties of materials find their origin in the orbital and spin motion of electrons, and how they interact with each other. The best manner to introduce the different types of magnetism is to describe how materials respond to an external magnetic field [27]. Typically, they are classified into diamagnetic, paramagnetic, ferromagnetism, ferrimagnetic, and antiferromagnetic materials.

Hysteresis loops are the most common characterization routines for studying the magnetic features of materials under the influence of an external magnetic field [27]. As shown in Figure 2, the saturation magnetization (*M_s_*), remanence magnetization (*M_r_*) and coercive field (*H_c_*) can be obtained from hysteresis loops. The first value corresponds to the magnetization when all the particle moments are aligned along the external applied magnetic field and is obtained by a linear fit of the high field zone of the hysteresis loop curve. The second corresponds to the residual magnetization when the applied magnetic field is removed. Finally, *H_c_* corresponds to the magnetic field that must be applied to cancel the sample magnetization. Magnetic (nano)particles are composed of magnetic domains, which are regions with uniform magnetization separated by domain walls and which help in reducing the magnetostatic energy. Among magnetic materials, ferromagnetic and ferrimagnetic are the most chosen systems for preparing magnetic self-healing composites since they exhibit spontaneous magnetization in the absence of a magnetic field due to the full or partial parallel alignment of their magnetic moments. If the volume of the particle is small enough, it can be composed of a single magnetic domain (see Figure 2a). Due to their small volume, single-domain particles usually present a superparamagnetic behaviour, which means that thermal energy is sufficient to spontaneously change the magnetization of the particles, leading to an absence of remanent magnetization in the absence of a magnetic field [28]. However, in the presence of a magnetic field, there will be a net statistical alignment of the magnetic moments analogous to a paramagnet but with a higher magnetic moment, which can be up to 10^4^ times larger than that of an atomic paramagnet. To name some examples, superparamagnetic iron oxide nanoparticles (SPIONs) composed of Fe_3_O_4_ or γ-Fe_2_O_3_ have been widely investigated for applications in health, e.g., as magnetic resonance imaging (MRI) contrast agents, drug delivery systems or antennas for magnetic hyperthermia, due to their low toxicity and good magnetic performance [1,2,3,4].

**Figure 2 molecules-27-03796-f002:**
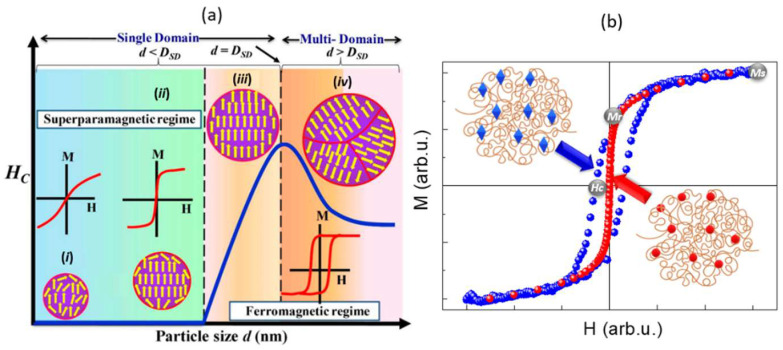
(**a**) A scheme showing the effects of the particle size with variations in *H_c_* and *M_s_*, from small superparamagnetic nanoparticles to multidomain domain magnetic (nano)particles. (i) Nanoparticles smaller than 3 nm show superparamagnetic behaviour with *Hc* = 0, and moderate values of *M_s_* due to the large contribution of paramagnetic atoms on the particle surface. (ii, iii) As the particle size increases, longer relaxation times are observed, stabilizing the blocked state and increasing the *H_c_*. (iv) Lastly, above the critical size for a single-domain formation, the particle is multidomain, which leads to a decrease in *H_c_*. (**b**) Hysteresis loops of hard (left) and soft (right) self-healing magnetic composites at the same temperature. The saturation magnetization (*M_s_*), the remanence magnetization (*M_r_*) and the coercive field (*H_c_*) are indicated in grey solid spheres. The figure is reprinted under an open access Creative Commons CC BY 4.0 license [29].

Magnetic (nano)particles can also be classified as soft or hard depending on their responses to an external magnetic field [4]. Soft-magnetic (nano)particles are generally composed of iron-nickel alloys, iron-silicon alloys, soft-ferrites, etc. They are characterized by showing low coercivity and high permeability. Consequently, they cannot retain *M_r_* after the external magnetic field is removed (see Figure 2b). In addition, when an external magnetic field is applied, these systems tend to align due to the strong magnetic interactions induced by their high permeability. The main inconvenience of soft magnetic self-healing composites is their lower programming flexibility due to the lack of magnetization retention in the absence of a magnetic field. Soft magnetic self-healing composites can typically be found in magnetic hydrogels and elastomers. Hard magnetic (nano)particles show, on the other hand, a high coercivity and a large hysteresis loop area, but a large magnetic field is required to reach *M_s_*. Typically, hard magnetic self-healing composites contain particles composed of neodymium-iron-boron, samarium-cobalt and hard ferrites, among others. They are characterized by good material programmability and consequently allow for more complex shape transformations.

## 3. State-of-the-Art Fabrication Methods of Magnetic Self-Healing Composites

Self-healing magnetic composites can be prepared by incorporating magnetic (nano)particles containing ferromagnetic or ferrimagnetic materials in self-healing polymer matrices such as soft elastomers or shape memory polymers [4]. The composition, content and surface functionalization of incorporated magnetic (nano)particles are key factors that determine the performances of nanocomposites. The utilization of hard magnetic ferrites (nano)particles in robotics has attracted much attention due to their low magnetic permeability under strong magnetic fields, boosting material programmability [30,31,32]. The magnetic response of magnetic self-healing nanocomposites usually scales up with the particle content. However, particle aggregation and phase separation are usual bottlenecks limiting particle loading [4]. Finally, the grafting of inorganic and organic molecules on the particle surface can introduce new functionalities to the final material and enhance the particle dispersion within the polymeric matrix [32,33,34,35,36].

So far, two main strategies have been adopted to induce the self-healing of nanocomposites: autonomous and nonautonomous damage healing [37]. The nonautonomous approach relies on external interventions, such as an inductive heating effect under an external stimulus, e.g., induced by an alternating magnetic field for magnetic (nano)particles or near-infrared light for light absorbers [38,39,40,41,42]. The local heat produced from these processes enables various types of self-healing strategies, such as melting of a thermoplastic phase, facilitating chemical bond exchange or activating reversible reactions. This approach has yielded magnetic self-healing composites with good self-healing and mechanical properties, but it shows the limitation of requiring the application of an external stimulus. Autonomous healing does not require any human intervention and relies either on an embedded healing agent that is released autonomously upon mechanical damage to the reservoir it is stored in or on the intrinsic self-healing potential of polymeric matrices involving dynamic covalent bonds or noncovalent interactions that can reform at the application temperature. Dynamic covalent chemistries include Diels-Alder (DA) reactions, disulfide, imine and boron-based bond exchange [35,43,44,45,46]. Noncovalent chemistries refer to hydrogen bonds, ionic interactions, metal-ligand bonding, electrostatic interactions and π-π staking [30,32,47,48,49,50]. It is, however, worth mentioning that the distinction between these two strategies is not always clear and that many intrinsic self-healing systems need a temperature increase to reach a fully healed state, which for example can be induced by magnetic inductive heating as a combination of these two strategies [45]. Table 1 summarizes the state-of-the-art fabrication methods of magnetic self-healing nanocomposites. The material geometry and composition determine the final functionalities of magnetic self-healing composites. In a general view, the mixing and dispersion of magnetic (nano)particles in a polymeric matrix can be used to fabricate simple 2D and 3D geometries. Additive manufacturing methods allow for the fabrication of complex structures in short times but at a higher production cost.

**Table 1 molecules-27-03796-t001:** The most utilized fabrication methods for magnetic self-healing composites together with their pros and cons.

Method	Pros	Cons	References
Direct mixing and dispersion of magnetic (nano)particles in a polymeric matrix	Simple fabrication Scalability Low cost	Simple 2D-3D geometries NP agglomeration	[30,39,40,41,42,45,49,50]
Self-assembled magnetic (nano)particles composites	Good control over the particle aggregation state Incorporation of additional functionalities	Complex fabrication procedure	[32,33,34,36,46,48,51,52,53]
Additive manufacturing	Fast fabrication Multi-material printing High resolution	Complex control and high cost	[33,45,46,49]
AC magnetic field	Fast fabrication	AC magnetic fields are required	[54]
Spray deposition	Fast fabrication Easy performance	Limitation in building geometries	[38]

### 3.1. Direct Mixing and Dispersion of Magnetic (Nano)Particles in a Polymeric Matrix

This strategy involves the mixing and dispersion of magnetic (nano)particles in organic or aqueous media under thermal or ultrasonic stimulation followed by sample casting or moulding. Mechanical and magnetic features of the resulting materials will depend on the selection of the polymeric matrix, the particle features and their compatibility/interactions. This strategy has been explored for self-healing magnetic composites ranging from hydrogels to elastomers. 

Magnetic self-healing hydrogels combining Fe_3_O_4_ NPs and poly(vinyl alcohol) (PVA) have been reported by Chen et al., who describe an effective manner of embedding Fe_3_O_4_ in PVA by moulding polymer aqueous solutions with particle contents ranging from 2 to 10 wt% at low temperature [50]. As the particle content increased, an enhancement of the tensile strength from 0.13 MPa to 0.32 MPa was observed because of the formation of a physical particle network of the large, stiff Fe_3_O_4_ particles in a soft polymer matrix. Interestingly, the self-healing efficiency increased up to 5 wt% of Fe_3_O_4_ and then diminished, probably due to the formation of large particle agglomerates, yielding a poor interfacial interaction with the polymer matrix and reducing the polymer chain mobility required for efficient damage healing.

The incorporation of magnetic (nano)particles in thermoreversible gels results in a material that undergoes a reversible gel transition when it is heated by the application of an alternating magnetic field. The ability of electromagnetic fields to penetrate thick composites combined with the ability of such materials to be repeatedly heated and healed could prove particularly interesting in applications where large polymeric objects are subjected to repeated stresses such as wind turbines or helicopter rotors. Adzima et al. described an elegant methodology for preparing magnetic self-healing gels by the sonication of trifuran (pentaerythritol propoxylate tris(3-(furfurylthiol) propionate) with (1,1′-(methylene-di-4,1-phenylene) bismaleimide using chromium (IV) oxide (CrO_2_) NPs from 0.1 to 10 wt% [45]. The as-prepared magnetic gel showed a reversible gel transition after being incubated under an alternating magnetic field of 15 mT and 390 kHz for 150 s. Note that the steady-state temperature reached by the polymer composite under the mentioned conditions increases with the particle content from 120 °C for 0.1 wt% to 146 °C for 10 wt%.

Figure 3 shows a room-temperature self-healing magnetic soft elastomer made of a soft poly(dimethylsiloxane) (PDMS) matrix mixed with 20 nm-Fe_3_O_4_ NPs as functional magnetic nanofiller with wt% ranging from 10 to 25% [30]. The preparation consisted of the mixing of an aqueous/methanol solution of PDMS-COOH with Fe_3_O_4_ NPs by sonication, followed by moulding and drying at 120 °C for 12 h, and finalized by several hot-pressing steps. The optimum content of the magnetic filler was determined to be 15 wt% with a tensile strain of 400% and a self-healing efficiency of 62.2% at room temperature. Another interesting example was shown by Hohlbein et al. [42]. In this work, Fe_3_O_4_ and CoFe_2_O_4_ NPs with spherical, cubic and rod shapes were embedded in the ionomeric elastomer PZn-52. The fabrication was performed by mixing the elastomer and the magnetic NPs under sonication followed by moulding and drying in vacuo for 72 h. The best healing was found for elastomers containing multidomain cubic Fe_3_O_4_ NPs under an alternating field of 250 kHz (31.5 kAm^−1^). The proposed materials were successfully exploited as magnetic soft actuators.

While casting and moulding methods are the most popular choices for the preparation of magnetic self-healing composites, they exhibit strong limitations concerning magnetic (nano)particle aggregation and sedimentation. This is a key issue for the fabrication of most polymer nanocomposites, where magnetic composites find the extra disadvantage of magnetic dipole interactions when the particles are magnetized [55]. Self-healing materials are usually built up from monomeric mixtures with a low molecular weight that exhibit low-viscosity, liquid-like behaviour. Upon polymerization, the reversible polymer network structure and related viscoelastic properties build up. Mixing magnetic (nano)particles in the initial unreacted monomer mixture, under reprocessing upon degelation or in the presence of a solvent, intrinsically leads to strong magnetic (nano)particle interactions and the sedimentation of aggregates or the formation of strong percolated structures in these low-viscosity systems. These are known to take a predominant role in the rheological behaviour and mechanical performance of the composites [56]. This is detrimental to some applications for which finely dispersed magnetic NPs are required, such as MRI or inductive heating. Hence, alternative synthetic methods or composite (re)processing are required to shatter the agglomerate particle’s state and keep the dispersion stable until network (re)formation. Fabrication methods that can improve magnetic (nano)particle dispersion could be divided into two families:Physical methods: enhance the particle deagglomeration by applying high shear forces that can break down the particle agglomerates to nanoscale dispersed particles. This includes well-known top-down manufacturing methods such as twin-screw extrusion [57,58], injection moulding [59], roll and ball milling [60,61,62] and ultrasonic vibration [63,64]. The exerted high shear forces are capable of reducing the aggregated particles sufficiently and homogeneously dispersing them. However, there is the great inconvenience that dissociative covalent networks or supramolecular networks rely on the narrow thermal region of the gel transition linked to an abrupt viscosity drop. Thus, precise control of the reaction conversion needs to be maintained to ensure a sufficiently high viscous medium to avoid particle re-aggregation and sedimentation upon curing. With reversible polymer networks, this can be achieved upon thermal dissociation at a well-defined temperature close to the gel transition. In addition, a monodisperse composite cannot be achieved under these conditions as shear forces are not high enough to separate the clusters at smaller size scales.Chemical methods that can deal with the dispersion issue from a composition perspective: Hindering magnetic (nano)particle interactions (mostly magnetic dipole-dipole interactions, hydrogen bonding and van der Waals interactions) by designing proper formulations has been proven to enhance (nano)particle distribution. This can be approached from either the magnetic (nano)particles (surface functionalization, magnetization state, etc.) or the polymer matrix state (polymer (nano)blends [65,66,67,68], the grafting of diverse functional groups [48,53,69], etc.). Additionally, the incorporation of stabilizers [70] or performing the magnetic (nano)particle synthesis of the polymer matrix in situ at a suitable crosslinked state [55] can help further this aim.

Dynamic covalent chemistries blur the line between physical and chemical methods given their reversible nature and resulting processability upon the application of an external trigger [71]. Supramolecular networks and dissociative covalent chemistries undergo a shift in their equilibrium state caused by an external perturbation, e.g., thermal heating or light irradiation. In addition, high shear rates can also lead to a mechanochemical dissociation of dynamic covalent crosslinks since these bonds have been found to be the weakest for most of the proposed reversibly crosslinked materials [69]. This enables physical processing methods to be adapted and employed as a consequence of the thermally and mechanically induced reversible chemical changes of the self-healing polymer networks. Analogous strategies are well-known for thermoplastics. While traditional thermoplastics benefit from higher melt strength and extensional rheology or fast crystallization kinetics, following their high molecular weight and thermophysical properties, reversible polymer networks rely on reversible chemical (de)polymerization reactions with associated molecular weight decrease and chemical reaction kinetics. The design of more complex and efficient reversible crosslinked polymers or blends, or the use of exchange reactions that do not show a net change in crosslink density, are potential routes towards the rapid and efficient manufacturing of magnetic self-healing composites [72].

### 3.2. Engineering of the Particle Surface towards the Self-Assembly of Magnetic Self-Healing Composites

The grafting of organic molecules on particles surface represents a step forward towards the development of intrinsically magnetic self-healing composites presenting a better compatibility between the magnetic (nano)particles and the polymer matrix. Functionalization strategies are divided into ligand addition and ligand exchange. In the former strategy, amphiphilic molecules interact with the part of the surfactant attached to the particle surface, forming a double layer that covers the particle surface. In the latter strategy, the ligands attached to the particle surface are replaced with new functional compounds by exchange reactions. The new compounds have typically one functional group for binding to the particle surface via strong chemical bonds, e.g., epoxy, catechol, amine, phosphate, and another terminal group that enables the dispersion of magnetic (nano)particles in the polymeric matrix.

Surface-modified magnetic (nano)particles with organic groups capable of thermoreversible cross-linking are versatile building blocks for magnetic self-healing nanocomposites with improved lives. Within this context, exploiting the reversible Diels-Alder (DA) reaction has been regarded as a useful strategy [37,47,48]. Schäfer et al. developed a system based on 20 nm Fe_3_O_4_ NPs that were co-functionalized with alkyl phosphonic acids terminated or by a methyl group or a functional group that can be involved in DA reactions as a furan or a maleimide [46]. The number of DA functional groups was systematically changed from 0 to 100% in 25% steps. The furan-grafted Fe_3_O_4_ NPs were chemically crosslinked using a bismaleimide to form polymer networks with magnetic crosslinking centres. Scratch healing was demonstrated at 120 °C for 1 h, as a significant amount of mobility was required to fill and heal the scratch damage volume. Counter-intuitively, the dilution of the DA reactive groups on the NP surface was shown to lead to lower conversions of the reactive groups into DA adducts. This was explained by steric hindrance. Increasing the spacer length of the grafting agents from propyl to decyl led to a 3- to 4-fold increase in the reaction conversion. Another interesting approach was recently shown by Muradyan et al. where 20 nm Fe_3_O_4_ NPs were functionalized with a self-healing matrix composed of the low-cost commercial monomers acrylamide and n-butyl-acrylate using a graft-from approach [32]. The resulting magnetic self-healing nanocomposite reached a Young’s modulus of 70 MPa and extensibility of over 500% while displaying self-healing behaviour at room temperature. 

Magnetic self-healing thermoplastics have been reported with functionalized magnetic (nano)particles of different sizes and compositions. For example, Yoonessi et al. reported the fabrication of magnetic polystyrene nanocomposites by compression moulding at 240 °C for 20 min of polystyrene powders containing 20 nm core-shell (CoFe_2_O_4_-MnFe_2_O_4_) NPs with particle content ranging from 3.5 to 10 wt% [34]. When the magnetic polymer nanocomposites were placed in an AC magnetic field, they showed high heat generation due to Néel relaxation and hysteresis of the core-shell magnetic (nano)particles in the solid state. The heat generation resulted in the fusing and healing of large-scale cracks. The maximum temperature generated was for the polymer with 10 wt% of particle content. Another interesting manner of boosting the magnetic hyperthermia efficiency is by increasing the Curie temperature of the particles. Ahmed et al. reported the preparation of magnetic polyethylene vinyl acetate polymer, functionalizing 20 nm Mn_x_Zn_1−x_Fe_2_O_4_ NPs with poly(ethylene-co-acrylic acid) to avoid agglomeration during the processing of the composite [39]. The resulting magnetic (nano)particles were added to the polymer by solution casting with final contents from 10 to 16 wt%. The magnetic self-healing was enhanced by tuning the Curie temperature by adjusting the Mn:Zn ratios, which ranged from 0.9Mn:0.1 Zn to 0.7Mn:0.3 Zn. The magnetic nanocomposite containing Mn_0.8_Zn_0.2_Fe_2_O_4_ reached superior recovery to that of other compositions, as higher temperatures were reached when it was placed in an AC magnetic field, showing better self-healing efficiency than other compositions.

The direct polymerization of ligands onto the particle surface is limited by the need for compatibility between the coating material and the reaction conditions, e.g., solubility, pH and temperature. To overcome this issue, the coating and polymerization processes can be split into two or more steps. Magnetic hydrogels with chitosan- Fe_3_O_4_ NPs as crosslinking points and telechelic difunctional poly(ethylene glycol) (DF-PEG) as the polymeric matrix were built using a dynamic covalent Schiff-base linkage between the NH_2_ groups on chitosan and the benzaldehyde groups on the PEG chain ends [47]. The as-prepared magnetic hydrogel showed self-healing at room temperature without additional stimulus and excellent biocompatibility with HeLLa cells. Another interesting example was shown by Qiaochu Li et al., where Fe_3_O_4_ NPs were functionalized with catechol groups inspired by the additive chemistry in mussel byssal threads (see Figure 4) [36]. Fe_3_O_4_ NPs functionalized with polyethylene glycol molecules that ended with a carboxylic acid were mixed with 4-arm catechol-terminated polyethylene glycol. As a result of the interaction between the catechol moieties and the Fe(III), magnetic gels could be obtained at particle concentrations of 1 vol % and above. The grafting or conjugation of different ligands on the particle surface can contribute to the integration of several functionalities into the final material. Recently, Kai Liu et al. showed an ultrasoft self-healing magnetic hydrogel with conductive and magnetic features in a PVA polymer using MnFe_2_O_4_ NPs and nano-fibrillated cellulose functionalized with polyaniline as the substrate [48]. The maximum saturation magnetization and conductivity of the materials were 5.22 emu·g^−1^ and 8.15 × 10^−3^ S·cm^−1^, respectively. Additionally, the PVA hydrogen showed good self-healing property, as it could be completely healed after the pieces of the hydrogel were put together for several minutes at room temperature. 

The synthesis of magnetic self-healing polymers using SPIONs was reported in three steps [53]. Magnetic (nano)particles were first activated with the agent KH-570 to enable greater interface compatibility. The resulting activated particles were then functionalized with NVP and DVP monomers and, finally, reacted with the agent TETA and a fatty acid dimer. The superparamagnetic polymer reached 74.3% healing efficiency in 1 h at room temperature and 77% when it was incubated under an alternating magnetic field.

### 3.3. Additive Manufacturing

Since its invention in the 1980s, additive manufacturing, popularly named 3D printing, has attracted much attention owing to its revolutionary ability to rapidly manufacture hierarchically complex geometries in a simple and customizable manner [73,74,75]. This fabrication method is used to construct an object layer by layer. By controlling the printing direction, structures with complex geometries and magnetization distributions can be fabricated for the desired application. Extrusion-based 3D printing is one of the most recent and attractive methods for manufacturing self-healing hydrogels, as optimal mechanical properties along with their self-healing ability facilitate gel stacking and interfacial bonding during the fabrication of 3D geometries. Ko et al. showed the versatility of the extrusion-based printing method in the fabrication of a magnetic self-healing hydrogel formed from the polysaccharides glycol chitosan and oxidized sodium hyaluronate in the presence of SPIONs without the addition of any chemical crosslinkers [44]. The printing was performed with a 25-gauge needle as a nozzle, and the ferrogel printing speed varied from 3 to 7 mm/s. The prepared hydrogel showed nonsignificant toxicity with ATDCS cells and a maximum self-healing when the SPION content was ≥5 wt%. Gang et al. demonstrated the high performance of the 3D printing method in fabricating the complex structures (see Figure 5) of a magnetic self-healing hydrogel composed of a double network of chitosan-polyolefin matrix and bondable with 50 nm Fe_3_O_4_ NPs [47]. The magnetic hydrogel showed high strength (>2 MPa), self-healing behaviour after thermal treatment, good magneto-thermal and MRI properties and a good cytocompatibility on the L929 cell line after up to five days of incubation.

The fused filament fabrication (FFF) of the thermoplastic elastomers embedded within a silicone rubber matrix has also been reported for polycaprolactone and thermoplastic polyurethane using carbonyl iron particles with sizes ranging from 5 to 100 nm with a filler content from 40 to 80 wt% [31]. The prepared polymers showed excellent mechanical property recovery after five healing cycles as a consequence of the intermolecular diffusion provided by PCL when heated above the melting temperature. Additive manufacturing is shown as a promising processing approach to building magnetic self-healing structures. To date, there have only been a few attempts at the additive manufacturing of magnetic self-healing composites, focused primarily on filament printing techniques. Other common additive manufacturing techniques such as stereolithography (SLA), direct ink writing (DIW) and selective laser sintering (SLS) remain unexplored.

### 3.4. Alternative Fabrication Methods

#### 3.4.1. Polymerization Induced by an Alternative Magnetic Field

In addition to inducing self-healing, magnetic hyperthermia has been shown to be an effective manner of inducing polymerization in magnetic self-healing gels. This is because magnetic (nano)particles act as hot spots under alternative magnetic fields, igniting the polymerization process. The fabrication of a magnetic gel from β-cyclodextrin and N-vinylimidazole monomers in the presence of SPIONs under an external stimulus of 450 kHz for 30 s was evaluated by Yu et al. [54]. The fast polymerization process was induced by a frontal polymerization based on the propagation of a localized reaction zone through the whole system. The resultant supramolecular gel showed an autonomous self-healing efficiency of 95%. Interestingly, the healing kinetics could be increased up to 6 times when samples were treated under the magnetic stimulus. The supramolecular gel possessed high mechanical strength, with a stress intensity of 0.12 MPa and an elongation at break over 826%. In addition to network polymerization and self-healing enhancement, magnetic hyperthermia treatments can depolymerize thermoreversible networks by heating above the gel transition temperature (T_gel_), thus creating a low-viscous, reprocessable melt [45]. To the best of the authors’ knowledge, although it has been proven that this approach is feasible, no detailed evaluations on the reprocessing effectiveness are available in the literature.

#### 3.4.2. Spray Deposition Method

Spray deposition is a simple method of producing a broad range of magnetic nanocomposites. This method relies on the direct conversion of single-solution droplets generated by the automatization of a precursor in a one-step process. A precursor solution is usually sprayed into a hot wall reactor, where the solution evaporates. The solution droplets dry while airborne or in contact with the hot reactor wall and form the particles without any chemical reaction. Recently, Wei Li et al. developed a robust self-healing magnetic nanocomposite composed of a resin matrix formed by epoxy (EP) and polycaprolactone (PCL) together with Fe_3_O_4_ and SiO_2_ microparticles in two steps [38]. First, the EP/PCL mixed-resin matrix was prepared through spray deposition. Then, magnetic microparticles were deposited on the semi-wet resin matrix by gradient distribution under gravity, followed by subsequent solvent evaporation. The results showed that cutting cracks could be rapidly thermally healed under irradiation with near-infrared light. To summarize this section, Table 2 shows an overview of the different self-healing mechanisms and magnetic fillers evaluated for magnetic self-healing composites.

**Table 2 molecules-27-03796-t002:** The different self-healing mechanisms and magnetic fillers reported for the reported magnetic self-healing composite fabrication methods.

Fabrication Method	Self-Healing Mechanism	Magnetic Filler	References
Direct Mixing	Ionomer	Fe_3_O_4_, CoFe_2_O_4_	[41,42]
	Hydrogen bonds	Fe_3_O_4_	[32,35,51,52]
	Diels-Alder	CrO_2_	[45]
		Fe_3_O_4_	[56]
	Intermolecular diffusion	Fe_3_O_4_	[40]
NPs surface engineering and self-assembly	Metal-ligand complex	Fe_3_O_4_	[36,51]
	Hydrogen bonds	Fe_3_O_4_ Fe_3_O_4_@NVP-DVB	[32] [53]
	Boronic ester	MnFe_2_O_4_	[48]
	Diels-Alder	Fe_3_O_4_ Fe_3_O_4_@MWCNTs	[46] [35]
	Schiff base	Fe_3_O_4_ Fe_3_O_4_@SiO_2_	[52] [33]
	Intermolecular diffusion	Fe_2_CoO_4_@Fe_2_MnO_4_ Mn_x_Zn_1−x_Fe_2_O_4_	[34] [39]
Additive manufacturing	Hydrogen bonds/π-π stacking	Fe_3_O_4_	[49]
	Schiff base	Fe_3_O_4_	[43,44]
	Intermolecular diffusion	Fe_3_O_4_	[31]
External magnetic field	Host-guest interactions	Fe_3_O_4_	[54]
	Schiff base	CIPs	[76]
	Intermolecular diffusion	Fe_3_O_4_	[77]
Spray deposition	Intermolecular diffusion	Fe_3_O_4_@SiO_2_	[40]

## 4. Applications

Across the extensive range of well-established applications that use both magnetic soft materials and self-healing polymers, magnetic self-healing composites have proven to exhibit greater properties by combining both functionalities in a single system. A proper design and fabrication of dynamic reversible polymers together with a suitable dispersion of magnetic fillers within the matrix allow for the creation of objects with multi-stimuli responsiveness, high flexibility and toughness. Magnetic particles allow the response to magnetic fields, and thus, magnetic self-healing becomes an interesting choice for novel state-of-the-art technological or biomedical applications. The magnetically driven locomotion of soft actuators can be achieved by applying an external magnetic field. The resulting magnetic composites can be exploited, for example, for minimal invasive surgery or the manipulation of delicate objects [78]. Remote heating under the submission of a magnetic system to a high frequency alternating magnetic field can serve as an additional stimulus to boost the healing performance of self-healing materials. This is desired, together with polymers showing dynamic reversibility, for hands-free applications such as triggerable drug delivery or liquid surface manipulations with enhanced service time [45]. Moreover, coupling electric and magnetic fillers exploits the preparation of composites with multifunctional response [79,80]. Expeditious investigations of self-healing flexible electronics can take advantage of magnetism to promote their efficacy for technological applications such as flexible sensors, energy storage devices and electromagnetic interference (EMI) shielding systems. 

### 4.1. Actuators

The field of soft actuators is a fast-growing sector arising from the great advances in soft robotics and materials science. Inspired by nature, stimuli-responsive materials have substantially expanded their applicability as soft actuators by promoting their adaptation to the surrounding environmental conditions. Stimuli-responsive actuators are capable of responding to a diversity of external triggers such as heat, light, pH and electric or magnetic fields, leading to modifications of the system performance [81]. This includes the formulation of magnetic soft actuators whose locomotion or properties can be controlled under the application of a magnetic field. Fields such as magnetorheology [82], magnetostriction [83] and untethered robotics [84] have found ample opportunities in robotics due to their fast and reversible responses, versatility and applicability. Magnetic fields are of exceptional interest for this purpose as a consequence of their deep and efficient penetration throughout a broad range of materials, coupled with the quick reaction of magneto-responsive materials to dynamic changes in the magnetic field [85]. Separately, self-healing soft robotics is a new field that emerges as a solution to the inherent vulnerability of soft solids to harsh damage. This extends the performance lifetime of the soft robot part and improves the circularity and reliability as compared with conventional robotic systems due to the ability to heal micro- or macroscopic damages when exposed to a harsh actuation environment. In a recent review exploring the usefulness of self-healing materials for soft robotic applications, magnetic self-healing composites were proposed as promising candidates, using the magnetic field responsiveness for both actuation and promotion of the self-healing mechanism [14]. Thus, the combination of the aforementioned fields can pave the path to the formulation and application of novel materials that can exploit the advantages of both types of materials in a single system.

One of the most attractive characteristics of magnetic actuators is their remote controllability when submitted to magnetic fields. Supramolecular reversible networks have attracted great attention for this purpose because of their great stretchability, allowing controlled shape morphing even under the submission of mild magnetic forces. Cheng et al. combined dynamic linear backbones (disulfide bonds) and physicochemical interactions (hydrogen and coordination bonds) to synthesize hierarchically structured dynamic networks. During the synthesis, magnetic microparticles are incorporated to also provide a magnetic response to the system. The final composite exhibits a fast recovery, being able to sustain a 300% stretch after catastrophic damage when healed during 5 s at room temperature [86]. The proposed material is employed to synthesize a variety of different soft actuators, allowing the complex motions such as cargo transport (Figure 6a), multiterrain locomotion (Figure 6b) or climbing plants tendrils (Figure 6c). Composites resulting from reversible networks crosslinked at the particle surface are attractive candidates, suitable for exploitation for soft robotics. Muradyan et al. prepared a cost-effective magnetic self-healing nanocomposite for which the dynamic reversibility takes place at the particle surface via hydrogen bonding between amide functional groups. The synthesized composites exhibited an enhanced Young’s modulus of 70 MPa and extensibility of 500%, sufficient to be efficiently manipulated using magnetic fields, and a self-healing efficiency close to 50% after 5 h under ambient conditions [32]. Similarly, Shibaev et al. developed a dual-crosslinked hydrogel combining particle-polymer hydrogen bonds and dynamic borate crosslinks. The soft composite exhibits pH responsiveness via borate ion cleavage and efficient magnetic actuation [87]. In addition to magnetic fields, near-infrared-light (NIR) irradiation has been studied as a stimulus for magnetic self-healing elastomers using magnetic particles as photothermal conversion agents. Wang et al. developed multi-stimulus-responsive biomimetic soft microactuators based on a hierarchically structured metal-ligand-crosslinked supramolecular elastomer embedded with a 3D interconnected magnetic nanohybrid network [88]. The prepared microrobots exhibit great self-healing performance at room temperature and can be actuated upon NIR light and magnetic field exposure as illustrated in Figure 6d,e (reversible capture actuation and butterfly wing actuation, respectively).

Another well-known remote actuation method is the use of the shape memory effect. Shape-memory polymers can “memorize” a permanent shape conformation. When the material is subsequently manipulated and deformed, it undergoes a change from this temporary fixed state to the memorized permanent shape under the application of an external trigger, such as heat or light [89]. This actuation method has been combined with self-healing and magnetic properties to enhance its functionalities, resulting in so-called shape-memory-assisted self-healing (SMASH) [90,91,92]. Feng et al. reported a magnetic shape-memory self-healing elastomer with embedded spherical nanoparticles NPs with an average particle size of 12.3 nm prepared via in situ radical polymerization [93]. The hydrogen bond interactions between NPs and polymer chains provide the great self-healing response. Inspired by the interfacial adhesion chemistry of mussel threads, Du et al. designed a multiresponsive composite with self-healing capacity provided by metallo-supramolecular interactions at the surface of the magnetic particles when combined with cathecol-telechelic end-capped poly(ε-caprolactone). This material combines thermal, magnetic and light responsiveness for actuation as illustrated in Figure 7a [51]. Despite the manifested interest in the combination of shape-memory and self-healing properties for magnetic actuators, to date, no studies of SMASH have been reported for soft robotics. SMASH-based systems could exhibit interesting applicability for soft actuators, mostly due to using the combination of both magnetic actuation and shape memory to accomplish much greater motion degrees of freedom. Despite the clear advantages, SMASH capability is still restricted to narrow damage below 1 mm size. Cerdan et al. took another approach to obtain large damage recovery using magnetostriction of the broken surfaces (Figure 7b). This magnetically driven self-healing method enabled the repair of wide damage areas (several millimetres) by promoting the closure of the damage surfaces by an external magnetic field [56]. This approach is particularly interesting to develop soft robotics systems that perform their actuation duties in harsh, aggressive environments.

**Figure 7 molecules-27-03796-f007:**
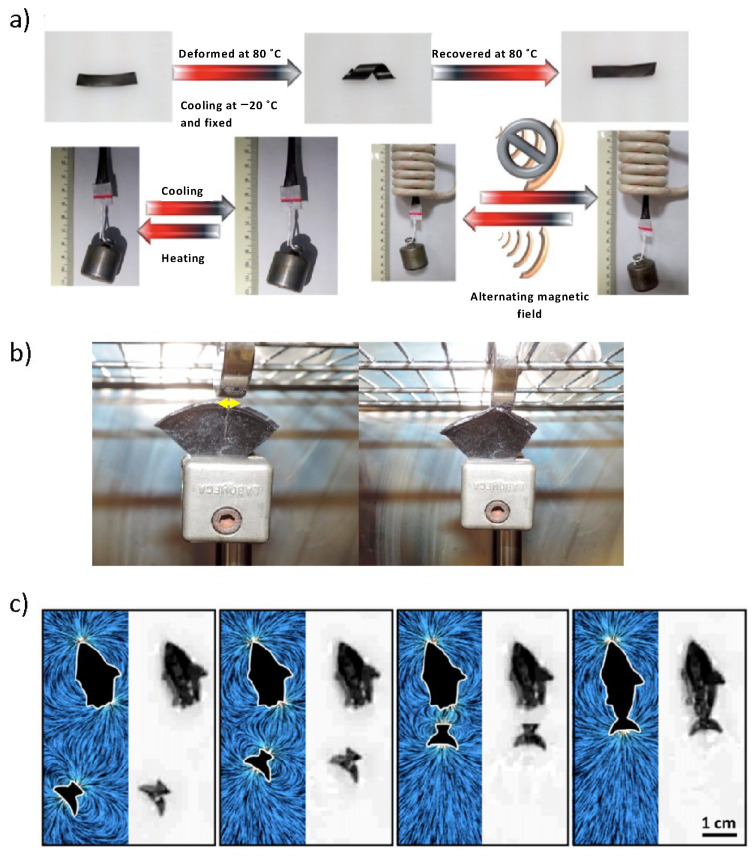
(**a**) Magnetic self-healing composite actuation using the shape-memory response under heating caused by a remote alternating magnetic field (adapted with permission from [51]. Copyright 2018, American Chemical Society); (**b**) magnetostriction-aided large damage healing (damage gap size of 4 mm–5 mm; adapted with permission from [56]. Copyright 2020, Elsevier; and (**c**) magnetic swimmer autonomously repaired “on-the-fly” using printed magnetic microparticles strips. Adapted with permission from [94]. Copyright 2021, American Chemical Society.

Finally, a novel approach to self-healing has been reported by Karshalev et al. to prepare microswimmers that can heal “on the fly”. These contain Nd_2_Fe_14_B magnetized microparticle strips within their structures, and as such, upon failure during swimming, the fractured parts can reorient and autonomously reattach restoring its propulsion behaviour (Figure 7c). This connection takes place in a moving system, showing great parts alignment and finding promising applicability in fields such as remediation or industrial clean-up [94]. While in this case the polymeric matrix did not recover its original mechanical properties, the microswimmers demonstrate a powerful approach to reattaching severed parts. The connection and alignment of damaged parts remains a major challenge in the self-healing of complex (robotic) systems.

### 4.2. Biomedical

Hydrogels have attracted great attention for biomedical applications in recent years. Their properties, similar to the extracellular matrix, such as high water content, biodegradability, suitable viscoelasticity, biocompatibility and porous framework, make them promising candidates for applications such as tissue engineering and bioimaging [95]. Tailoring factors such as concentration, molecular weight, functionality and stoichiometric ratio permit the tunability of their mechanical properties, which in turn influences their processing conditions and may guide cell fate processes [96]. The stimulus-responsive behaviour of reversible polymer networks has attracted much attention in addition to their self-healing ability. Dynamically crosslinked polymer networks allow novel routes towards the formulation of stimuli-responsive hydrogels that enable versatile adaptation to the surrounding environmental conditions. Conventional crosslinked hydrogels undergo several degradation mechanisms during their lifespan, such as mechanical failure and complex biodegradation controllability, that hinder their use in long-term applications. The ability to recuperate their mechanical properties under certain stimuli makes self-healing hydrogels better candidates for some in vivo biomedical applications that need sufficiently long mechanical durability to accomplish their tasks [97]. This is especially useful for minimally invasive operations performed by injectable hydrogels that perform, for example, as tissue-engineered scaffolds [98,99], wound dresses [100,101] or drug or cell delivery carriers [102,103]. Regularly, the extrusion of soft hydrogels through a nozzle requires mechanisms to undergo reversible sol-gel transitions such as pH or temperature (de)polymerizations or mechanical thixotropy to lower the system viscosity during the extrusion time [97]. The incorporation of dynamic crosslinks demonstrates self-healing hydrogels as excellent candidates to undergo these triggered transitions and heal the internal structure after injection. Furthermore, the incorporation of magnetic particles (mostly iron oxides due to their biocompatibility) within crosslinked hydrogels provides thermosensitivity when submitted to an alternating external magnetic field, finding use in minimally invasive therapeutic interventions and controlled drug release [104,105]. As a result, the formulation of self-healing dynamically crosslinked hydrogels embedded with magnetic particles is a little-explored field that can combine the advantages of the previously referred systems. A magnetic self-healing hydrogel was reported for the first time by Zhang et al. that dispersed Fe_3_O_4_ NPs within a dynamic hydrogel composed of chitosan and telechelic difunctional poly(ethylene glycol) [52].

Following this seminal work, several similar hydrogel matrix systems with embedded magnetic particles for biomedical applications have been reported. The most attractive application reported is their exploitation as chemotherapeutical agent thermocarriers. The resulting hydrogels enable targeted drug delivery and release by taking advantage of the hyperthermia provided by the magnetic particles. Xie et al. prepared a magnetic crosslinked hydrogel based on chitosan and telechelic difunctional poly(ethylene glycol) loaded with doxorubicin and docetaxel as a dual-drug-loaded system for breast cancer therapy. The formulated system exhibits better performance compared with single drug-loaded hydrogels, and the controlled drug release under an alternating magnetic field was also shown to improve the antitumor efficacy of the chemotherapy [106]. As a different matrix medium, Chen et al. exploited a Schiff’s base crosslinked hydrogel with dispersed magnetic gelatine macrocapsules that was also shown to be a promising candidate for drug delivery and soft tissue engineering [102]. Similarly, Li et al. formed crosslinked hydrogels from oxidized pectin/chitosan/nano γ-Fe_2_O_3_ via Schiff’s base reaction as a carrier for anticancer drugs [107]. Wang et al. investigated magnetic thermoreversible self-healing hydrogels where the self-healing properties emerge as a result of the electrostatic attraction between positively charged chitosan and negatively charged cellulose. The proposed scaffold showed a controlled in vitro drug release under an external magnetic field [108]. Charlet et al. showed magnetic self-healing hydrogels performing as excellent underwater glues. The proposed metal-coordinated telechelic hydrogels offer a wide tuning of their mechanical properties thanks to the possibility of crosslinking with a wide gamma of di- and trivalent ions. The resulting formulations are interesting candidates for biocompatible medical wound seals [109]. Most of the reported magnetic self-healing hydrogels are based on polysaccharides such as chitosan or gelatin. Other sources such as proteins have exhibited excellent self-healing efficiency and mechanical performance [110,111,112]. To our best knowledge, the combination of protein-based self-healing hydrogels with magnetic particles is still an unexplored field.

All the reported magnetic self-healing hydrogels have shown randomly dispersed magnetic particles involving an isotropic distribution. In contrast, the intrinsic properties of biological tissues as muscle or skin, such as enhanced mass transfer, surface lubrication, force generation and enhanced cell proliferation [113], arise from well-defined hierarchical, anisotropic microstructural arrangements. Consequently, some efforts have been carried out to overcome this drawback of traditional hydrogels by preparing anisotropic magnetic hydrogels. Nardecchia et al. prepared anisotropic self-healing magnetic hydrogels by submitting the uncured suspension to an external magnetic field during the polymerization step, as shown in Figure 8a. The resultant anisotropic material exhibits enhanced rheological properties when submitted to a magnetic field, emerging as a promising soft anisotropic material [76]. Zhang et al. prepared anisotropic magnetic self-healing material with both magnetothermal and photothermal responses. The magnetic particles absorb NIR irradiation, promoting the healing step through phase transformation. As can be observed in Figure 8b, the applied magnetic field can assemble the magnetic NPs as photonic crystals, exhibiting iridescent structural colours [77]. To date, no detailed studies on how the anisotropic arrangement of magnetic particles in self-healing hydrogels influences the healing properties are available. This would be of great interest since the particle assembly and packing are directly related to the polymer chains’ mobility and the dynamic groups’ rearrangement.

### 4.3. Stretchable Electronics

Flexible electronics are a rapidly growing field due to their capacity to experience high deformations while keeping their electric functionality. They find use in technological applications such as wearable electronics and sensors [114,115], health care systems [116] and flexible batteries and supercapacitors [117]. There are two predominant manufacturing strategies for the mentioned systems: intrinsic flexible and conductive polymers or hybrid composites containing a soft elastomeric substrate with interconnected conductive particles dispersed. The most popular choices among the latter include multiwalled carbon nanotubes (MWCNTs) and graphene due to their great mechanical properties, flexibility, high conductivity and anisotropy [118]. Thus, multiple conductive polymers as well as particle fillers embedded within soft elastic polymeric matrices have been explored and optimized for the desired applications. Magnetic particles have also been combined with flexible electronics to improve the material performance or enable an extra functionality as magnetoelectronic systems [119,120]. Thus, the formulation of self-healing, magneto- and electro- responsive systems drives the conceptualization, production and utilization of state-of-the-art, multi-stimuli-triggered smart materials. As a consequence of their inherent softness, stretchable electronics are also prone to lacerative damage or perforations that can limit their lifespans and utility. In addition, damage can yield the percolated conductive particle microstructures of filled polymeric composites, thus deteriorating the conductive response. In response, self-healing has been proposed as a solution to overcome the vulnerability of their conductive composites [121,122]; however, this process should restore not only the initial mechanical properties of the cured material but also the recombination of the conductive filler network. 

Thriving studies on conductive polymers applied to magnetoelectric self-healing composites have recently been proposed in the literature. Liu et al. developed an ultrasoft self-healing hydrogel with magnetic and conductive properties by dispersing MnFe_2_O_4_ NPs within a conductive polyaniline (PAni) matrix. NP aggregation was prevented using nanofibrillated cellulose (NFC) as a dispersant. Similarly, Zhao et al. prepared a biocompatible, remouldable composite hydrogel with conductive, magnetic and self-healing response by introducing a ZnFe_2_O_4_/MWCNT/Polypyrrole hybrid structure within a dynamic reversible hydrogel composed of a polyvinyl alcohol/borax double network. The hydrogen bonding between matrix and filler serves as an additional tool for mechanical strengthening [123]. He et al. embedded Fe_3_O_4_ NPs in a supramolecular network filled with imidazolium-based ionic liquids as a self-healing electronic sensor. Since the conductivity of ionic liquids is higher as the temperature increases, the exploitation of magnetic hyperthermia to improve the sensing performance was successfully confirmed in this study [124]. One promising route towards an enhanced self-healing performance was introduced by Bandodkar et al., consisting of the interactions among permanent magnetic particles (e.g., Nd_2_Fe_14_B microparticles) as an assisted healing driving force for self-healing graphitic inks. The formulated inks exhibit an impressive recovery of large damages in approximately 50 ms without any stimulus, restoring the conductivity and electrochemical properties [125]. Similarly, Guo et al. designed an Fe-doped liquid metal conductive ink combined with a degradable PVA substrate and adhesive fructose. The resulting multifunctional electronic system was able to perform as excellent light-emitting diodes (LED) able to be repaired upon magnetic field irradiation. This material was successfully applied on a crawling robot [126].

Supercapacitors are one of the most promising energy storage candidates as a consequence of the high-power density and large numbers of fast charging-discharging cycles that they can undergo. The introduction of flexibility in supercapacitors allows their use in a large number of portable and wearable lightweight consumer devices, finding applications in diverse fields such as medical, military or civilian fields [127,128,129]. Despite the encouraging opportunities, to date, magnetoelectric self-healing supercapacitors have been scarcely approached. As can be observed in Figure 9a, Qin et al. prepared a novel, magnetic, highly stretchable polyacrylamide hydrogel filled with Fe_3_O_4_@Au hybrid NPs. Polypyrrole NPs are built into the hydrogel network acting as electrodes. The Fe_3_O_4_@Au NPs and the hydrogel network are reversibly crosslinked at the gold surface with a disulfide bond-ended crosslinker via gold-thiolate bonds, reversibly triggered by remote photo-thermal, magneto-thermal and optical sources. These bonds provide a high multiresponsive healability, together with an areal capacitance of 1264 mF·cm^−2^ and restorability of 90% of initial capacitances over 10 healing cycles [130]. Huang et al. proved the healing and performance recovery of a yarn-based supercapacitor, which is particularly hard to heal due to its anisotropic wired geometry. In this work, magnetic yarn electrodes can be efficiently reconnected by magnetic attraction between the edges of the magnetic broken fibres, as shown in Figure 9b, while the mechanical properties recovery is attributed to a carboxylated PU supramolecular network electrode cover with abundant hydrogen bonding [131]. Triboelectric nanogenerators are a recently attractive choice for wearable devices due to their high energy conversion efficiency using mechanical stimuli [132]. However, the fact that they must endure mechanical forces to generate electronic carriers makes them highly prone to be damaged; thus, self-healing is a promising solution for the long-term loss of functionality of such systems. Xu et al. prepared a magnetic self-healing triboelectric nanogenerator. The healing performance is achieved by means of dynamic disulfide bonds with enhanced mechanical contact of the broken damage promoted by remote magnets (Figure 9). Both the mechanical and electric properties of the built-up electrodes are restored upon healing, and its high flexibility makes this system an interesting candidate for flexible electronics applications such as artificial skin [133]. The combination of triboelectric nanogenerators with supercapacitor systems can drive independent self-chargeable supercapacitors capable of generating their own current and storing it. Maitra et al. reported a triboelectric self-charging asymmetric supercapacitor with self-healing capacity. The proposed sandwich structure presents magnetic CoFe_2_O_4_ and Fe-decorated reduced graphene oxide that provides magnetic response to the system that can recover its conductive properties after damage thanks to its self-healing properties [125].

Finally, a very promising application of electromagnetic flexible polymeric composites relates to their use as electromagnetic interference (EMI) shields. Electromagnetic radiation is capable of interfering with other electronic devices, disturbing their functioning or even harming people if inadequately handled. Due to their light weight, low cost, corrosion resistance and processability, flexible polymeric conductive and magnetic nanocomposites have found a spot in the next generation of advanced EMI materials, outperforming the conventionally used metallic rigid shields [134,135,136]. For dynamic systems where flexible EMI shields are exposed to potentially aggressive environments, self-healing capacity can provide additional reliability to conventional protective systems. Dai et al. proposed a self-healing nanocomposite based on a dynamic covalent network crosslinked with imine reversible bonds, filled with a Fe_3_O_4_@MWCNTs hybrid filler as an EM wave-absorbing component. The proposed system exhibits decent EM wave-absorbing properties and recyclable performance when dissolving the material in an acidic solution [137]. Menon et al. initiated the polymerization of dissociative covalent networks by grafting reversible covalent bonds at the interface of MWCNTs and a polymeric crosslinker using Diels-Alder [138] and disulfide [139] reversible crosslinks. Magnetic NPs are added to the network to enhance the EMI shielding effect. In addition, the same research group reported a similar system but with shape-memory properties, promoting the SMASH capacity to enhance the healing performance [140]. Additional research is still needed on novel processing approaches in order to finely disperse magnetic and/or conductive nanofillers in the matrix to improve EMI radiation scattering and boost the EMI shielding effect while lowering the fillers concentration.

### 4.4. Slippery Surfaces

Droplet splashing and manipulation onto surfaces is a key phenomenon for applications such as microfluidic systems, microreactors and electronic refrigerators [141,142]. Tailoring the chemistry and topography can lead to the design of non-wetting surfaces with substantially diminished contact when exposed to liquids performing as, for example, excellent lubricants [143]. However, an inherent weakness of such types of complex surfaces is their vulnerability to too-aggressive forces, such as mechanical friction or high shear flow of viscous liquids. Thus, worn-out surfaces that possess healable properties to recover the non-wettability corresponding to the undamaged material state are highly desired. Magnetic self-healing slippery surfaces have been reported in the literature as an interesting route to recover both hydrophobicity and mechanical properties when exposed to aggressive damage. Jin et al. prepared slippery liquid-infused porous surfaces with magnetic and self-healing properties [144]. The proposed system possesses a dual-crosslinked polymer-particle network containing strong irreversible epoxy-amine crosslinks, together with reversible Fe-catechol sacrificial reversible crosslinks. The strong irreversible covalent crosslinks provide great material toughness, while the reversible bonds confer great stretchability and self-healing capacity. Finally, swelling silicone oil lubricant within the matrix provides excellent surface hydrophobicity and great slipperiness to water droplets, which are mostly recovered together with the mechanical properties upon several healing cycles when the material is damaged. Irajizad et al. reported, for the first time, a thermally triggered magnetic slippery surface with healing capacity based on a thin film of ferrofluid/oil supported on a ferrite magnetic tape substrate [145]. Droplets can be manipulated at the liquid-liquid interface under Marangoni convection generated by temperature gradients on the surface without any viscosity restrictions. The simplicity of the designed system overcomes the crucial drawbacks of other state-of-the-art slippery surfaces such as custom micro/nanofabrication or complex surface treatments. Table 3 offers a summary of the variety of applications in the literature reporting magnetic self-healing composites, together with the fabrication methods, self-healing mechanisms and magnetic fillers incorporated.

## 5. Perspectives

Self-healing materials are able to repair autonomously or under the action of a stimulus after damage and recover lost material properties, thus extending the lifetime of these materials and related products. Self-healing magnetic composites can recover both their mechanical properties and magnetic functional properties. In applications such as electronics, robotics and health care, derived systems are able to recover their sensory functions and actuation performance.

Magnetic (nano)particles have been introduced in thermally reversible polymer networks as a mechanism for remotely applying a thermal response using inductive or irradiative heating. Such a thermal stimulus can speed up the healing kinetics and increase the healing efficiency by activating the thermoreversible crosslinks or interactions. This thermal stimulus could also be used to achieve the thermal depolymerization of the reversible covalent polymer network for reprocessing or for reversible adhesion by the remote application of an alternating magnetic field; conversely, this stimulus could start a (frontal) polymerization reaction during synthesis or later during use. Similarly, inductive heating could also be used to activate other chemical or physical processes such as the shape-memory effect for improved damage closure or for actuation or locomotion.

The use of magnetic (nano)particles or components further improves the healing performance of the self-healing materials and structures, as it can bring the damaged surfaces back into contact, thus, closing the damaged volume. Soft magnetic particles can be embedded into a self-healing matrix to close large damage during the healing treatment in the presence of an external magnetic field. Contact between damaged surfaces and the alignment of the damaged areas remain major challenges in self-healing materials and composites. Therefore, magnetic self-healing composites provide an important perspective for successful damage realignment and closure, greatly improving the healing efficiency. Hard magnetic particles do not necessarily require the presence of an external magnetic field for the realignment of severed parts, as the inherent magnetic attraction of the severed parts results in their reconnection. If the magnetic attraction is strong enough to keep the severed parts attached, the functionality of the structure can be recovered even without an inherent self-healing ability of the matrix in which the particles are embedded.

Polymer processing has been thoroughly studied and optimized for a plethora of polymers and composites for decades. For the new classes of polymer networks, i.e., dynamic covalent networks and supramolecular materials, this field is much less understood, as most authors focus mostly on the synthesis and characterization of the self-healing materials rather than the (re)processability performance. The inherent reversible nature of the dynamic covalent chemistries used to create intrinsic self-healing polymers and composites blurs the line between thermal and chemical processing for covalently crosslinked polymer networks. Reprocessing can often be performed using the same (thermo)mechanical methods used for thermoplastic composites, extending the reprocessability to covalent polymer network composites. The additive manufacturing of chemically crosslinked polymer composites has been very challenging and mostly been available through vat polymerization, with typical issues combining the photopolymerization of low viscous resins with fillers. The thermal or other reprocessing of polymer network composites through, e.g., filament-based printing techniques, paves the way to a more facile way of the additive manufacturing of complex (self-healing) magnetic composites. The inherent reversibility of the intrinsic self-healing polymer network formation further enables the recycling of the self-healing magnetic composites, which is not generally available for covalently crosslinked polymer composites. The recycling of magnetic composites needs to consider the structure and morphology formation to achieve homogeneously isotropic or magnetic field-induced anisotropic structures, depending on the desired magnetoresponsive behaviour.

Most magnetic self-healing composite studies are focused on the self-healing aspects and the recovery of mechanical, magnetic and other functional properties of the material and of derived systems. Much knowledge and craftmanship from the field of magnetic composites have yet to find their way to the dynamic covalent and supramolecular networks that are used to create most intrinsic magnetic self-healing composites. Particle types; sizes, distributions and shape; and their dispersion throughout the polymer matrix present some of the many parameters for controlling and exploiting the magnetic properties of the formed composites and resulting possibilities for inductive heating and healing, large damage closure and actuation and sensing, among others.

## Figures and Tables

**Figure 1 molecules-27-03796-f001:**
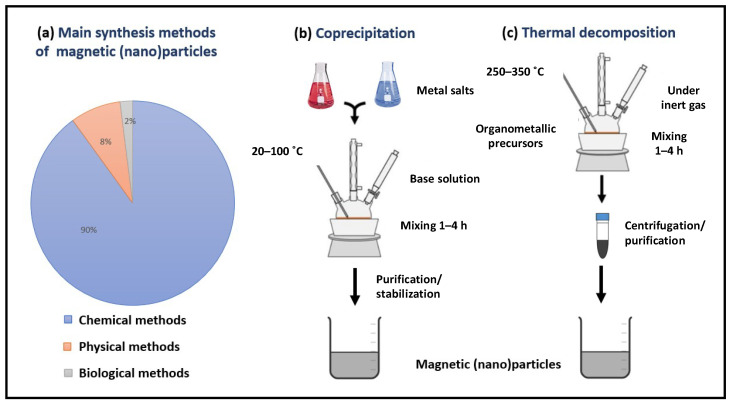
(**a**) Scheme depicting the main routes to synthesize magnetic (nano)particles. Diagrams for the preparation of magnetic (nano)particles employing (**b**) coprecipitation and (**c**) thermal decomposition approaches [22].

**Figure 3 molecules-27-03796-f003:**
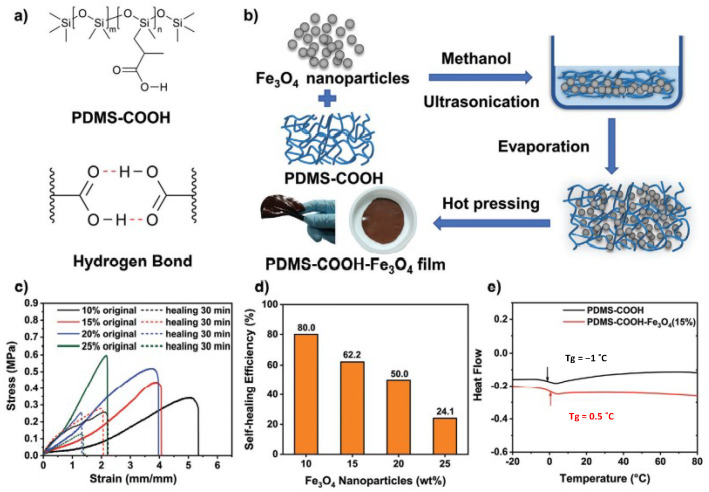
A sketch of the preparation and characterization of the PMMS-COOH-Fe_3_O_4_ polymer. (**a**) The chemical structure of the PDMS-COOH polymer. (**b**) the Schematics for the preparation process. (**c**) The stress-strain results for both the initial and healed polymers. (**d**) The self-healing efficiency of the PDMS-COOH at loading weights from 10 to 25 wt%. (**e**) DSC thermograms of PDMS-COOH and PDMS-COOH-Fe_3_O_4_ (15 wt%). Adapted with permission from [30]. Copyright 2021, John Wiley and Sons.

**Figure 4 molecules-27-03796-f004:**
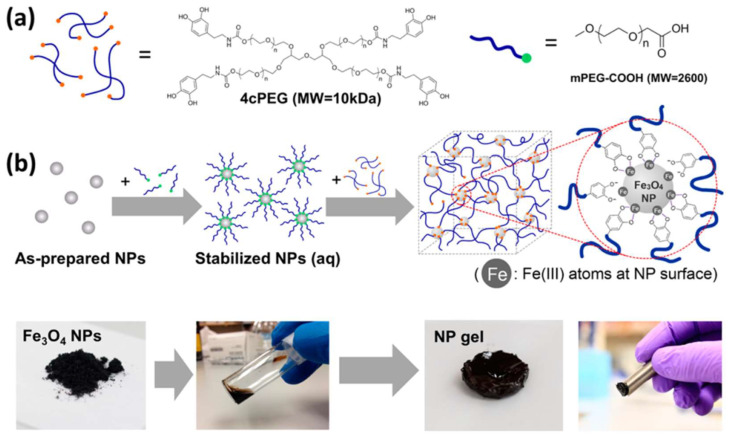
(**a**) The 4cPEG and mPEG-COOH structures of the polymers. (**b**) On the top is shown the preparation procedure for the Fe_3_O_4_ particles-cross-linked hydrogel. At the bottom are shown representative pictures at each state. From left to right: Fe_3_O_4_ dry powder, stabilized Fe_3_O_4_ NPs in aqueous dispersion before gel assembly, the self-standing solid hydrogel obtained after assembly with 4cPEG, and magnetic attraction of the resulting gel. Adapted with permission from [36]. Copyright 2016, American Chemical Society.

**Figure 5 molecules-27-03796-f005:**
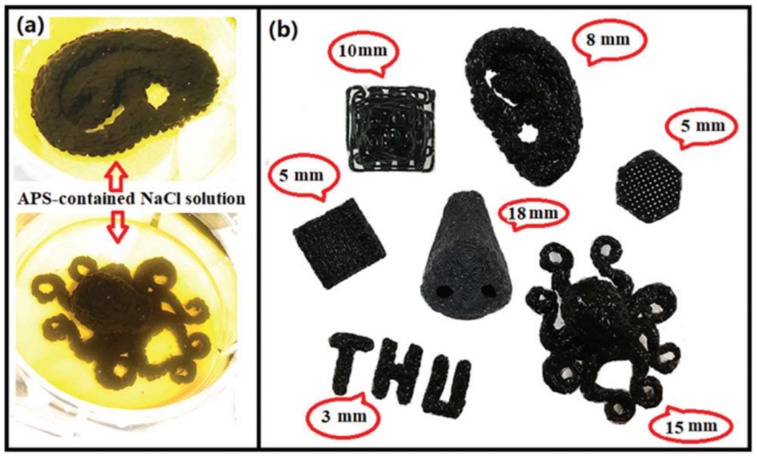
The geometries made by the 3D printing of the AAD-CS-Fe DN hydrogel containing 8 mg·mL^−1^ nano-Fe_3_O_4_. (**a**) A 3D model of the DN hydrogel in the pre-solution just after printing. (**b**) The 3D geometries made by the 3D printing. From left to right: Quadrangular pyramid, left ear, hexagonal prism, cuboid, nose, letters. The height of the corresponding models is shown in the red box. Adapted with permission from [30]. Copyright 2021, Wiley-VCH.

**Figure 6 molecules-27-03796-f006:**
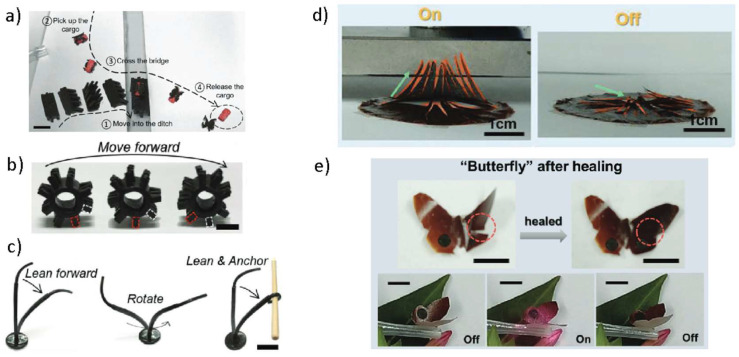
(**a**) A cargo transport task accomplished by a soft self-healing actuator under the presence of external magnetic fields; (**b**) a multiterrain carrier able to move forward using magnetic leg tilts; (**c**) magnetically controlled artificial soft tendrils. Adapted with permission from [86]. Copyright 2021, John Wiley and Sons; (**d**) mimosa-inspired magnetically driven capture-and-release actuation; and (**e**) synthetic butterfly wing healing under NIR irradiation and subsequent recovery of the vibratory actuation. Adapted with permission from [88]. Copyright 2019, John Wiley and Sons.

**Figure 8 molecules-27-03796-f008:**
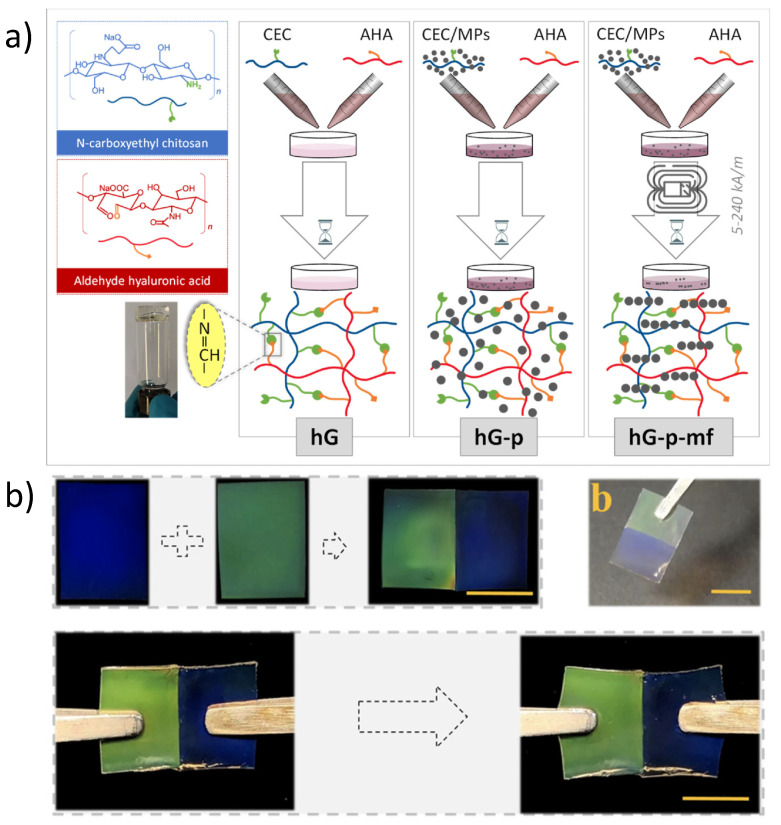
(**a**) The preparation of a magnetic anisotropic self-healing hydrogel composed of N-carboxyethyl chitosan and aldehyde hyaluronic acid crosslinked by dynamic imine covalent bonds. Adapted with permission from [76]. Copyright 2018, American Chemical Society. (**b**) The welding of two self-healing hydrogels with different structural colours. Adapted with permission from [77]. Copyright 2020, Elsevier.

**Figure 9 molecules-27-03796-f009:**
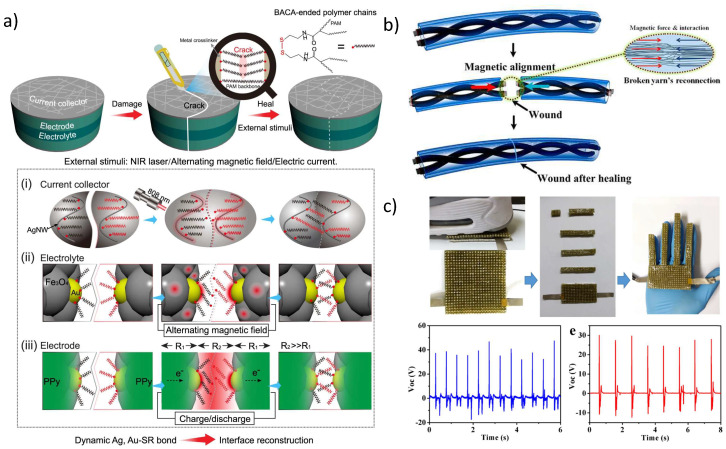
(**a**) A schematic representation of the interface reconstruction process of a magnetic self-healing supercapacitor induced by NIR irradiation [130]. (**b**) The healing of a yarn-based self-healing supercapacitor with excellent reconnectivity promoted by magnetic interactions. Adapted with permission from [131]. Copyright 2015, American Chemical Society. (**c**) An artificial skin based on a triboelectric nanogenerator, showing excellent conductive property recovery when the material is submitted to catastrophic damage. Adapted with permission from [133]. Copyright 2017, Elsevier.

**Table 3 molecules-27-03796-t003:** The different self-healing mechanisms, magnetic fillers and fabrication methods reported for the evaluated magnetic self-healing application fields.

Application	Fabrication Method	Self-Healing Mechanism	Magnetic Filler	References
Actuators	Mixing	Disulfide bonds	Nd_2_Fe_14_B	[86]
		Boronic ester	CoFe_2_O_4_	[87]
	Surface eng.	Metal-ligand complex	Fe_3_O_4_@CNC	[88]
		Hydrogen bonds	Fe_3_O_4_	[93]
Biomedical	Mixing	Imide bonds	Fe_3_O_4_@DF-PEG-DF	[106]
		Schiff base	Fe_3_O_4_ γ-Fe_2_O_3_	[102] [107]
	Mixing + external magnetic field	Schiff base	CIPs	[76]
		Intermolecular diffusion	Fe_3_O_4_@PSSMA@SiO_2_	[77]
Slippery surfaces	Mixing	Intermolecular diffusion	Oil-based ferrofluid	[145]
Stretchable electronics	Mixing	Hydrogen bonds	Fe_3_O_4_@SiO_2_ Magnetic cubes	[124] [133]
		Metal-ligand complex	Fe_3_O_4_@MoS_2_	[140]
		Imide bonds	Fe_3_O_4_@MWCNTs	[137]
		Diels-Alder	Fe_3_O_4_@rGO	[138]
		Disulfide bonds	Fe_3_O_4_@MoS_2_@rGO	[139]
	Surface eng.	Hydrogen bonds	ZnFe_2_O_4_@MWCNTs	[123]
		Disulfide bonds	Fe_3_O_4_@Au	[130]
	Additive manufacturing	Intermolecular diffusion	Nd_2_Fe_14_B Fe-GaIn	[125] [126]
	Electrodeposition	Hydrogen bonds	Fe_3_O_4_@Stainless Steel	[131]

## Data Availability

Not applicable.

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
