# Peer review of "Magnetic Self-Healing Composites: Synthesis and Applications"

_molecules, 2022, doi:10.3390/molecules27123796_

Round 1

Reviewer 1 Report

I found the paper well organized and comprehensively described. The work has scientific merit and deserve to be published. There seems to be some typos such as ; PMMS instead PDMS etc must be rechecked and corrected.  Overall, I found the paper acceptable

Author Response

We thank the referee for their appraisal of our work and the detailed revisions that enabled us to further polish the manuscript. We have thoroughly reviewed the manuscript to correct the mentioned typos and improved the English language of the manuscript. The referee can find the few requested changes in the text highlighted in yellow in the revised version of the manuscript.

Reviewer 2 Report

The authors presented a detailed review study of the synthesis and applications of self-healing magnetic composites, including a good summary of the most recent fabrication methods and an informative introduction to the well-established applications in the fields of actuators, biomedical, stretchable electronic and slippery surfaces. In general, the paper is well written and covers a thorough literature review of the relevant research studies in the field. I think the paper fits well with the scope of Molecules Journal, and it can be accepted after proof reading.

Author Response

We thank the referee for their appraisal of our work. In the present version of the manuscript, the referee could find the new parts highlighted in yellow and a more polished English version writing.

Reviewer 3 Report

The authors had covered a very interesting area in the present review article “Magnetic Self-Healing Composites: Synthesis and Applications”.

However, the link between magnetic materials and self-healing materials is missing in all the sections.

Thus, the review article requires a major revision before acceptance.

Author Response

We thank the referee for their positive opinion of our work. The link between the magnetic materials and the self-healing materials is not missing throughout all the sections, since all sections cover magnetic self-healing materials. These are materials that are both magnetic and self-healing. The link is inherent. We regret that this was not clear enough to the referee and we have made some adjustments to the starting and concluding paragraphs of each section to highlight this.

Reviewer 4 Report

Authors present paper "Magnetic Self-Healing Composites: Synthesis and Applications"

The topic of the review is important for various applications. There are many interesting examples of magnetic composites. Thank you for an exciting review.

I have only some comments to improve the review.

1) From the biomedical section 4.2 I don't clearly understand why we need self-healing polymer for drug delivery? Your examples look like it is just a surfactant coated magnetic nanoparticles or gel. And the surfactant is only used for the stability. If you can present interesting examples or your ideas, it will be great for biomedical area.

Also, I haven't seen is it possible to use for such systems biopolymers as proteins, etc.?

2) I highly recommend summarizing various applications with an exiting examples as a Picture somewhere in the review.

3) It will be very good also summarize the synthesis methods of magnetic self-healing systems and widely used polymers and surfactants in one or several Tables. 

Author Response

The topic of the review is important for various applications. There are many interesting examples of magnetic composites. Thank you for an exciting review. I have only some comments to improve the review.

1) From the biomedical section 4.2 I don't clearly understand why we need self-healing polymer for drug delivery? Your examples look like it is just a surfactant coated magnetic nanoparticles or gel. And the surfactant is only used for the stability. If you can present interesting examples or your ideas, it will be great for biomedical area.

The authors appreciate this observation. The use of self-healing polymers for biomedical applications resides in their role as a carrier. It is not directly related to the drug release itself but about the medium where these drugs are encapsulated and dispersed. The fact that they can undergo changes in their crosslink density (hence mechanical properties) and recover them afterwards makes them very interesting formulations as injectable hydrogels. These hydrogels can lower their crosslink density upon injection by applying a certain external trigger and recover their crosslink density when injected in vivo in a minimally invasive way. In addition, mechanical damage is a usual issue for regular hydrogels where forces are applied while they are, for example, proliferating cells as tissue scaffolds, due to for example human motion. The use of self-healing material suits hydrogels for long-time lasting applications. All this information has been thoroughly expanded and clarified in the main text, together with a better applications exemplification in fields such as drugs/cells carriers, tissue regeneration scaffolds and wound dressing.

Also, I haven't seen is it possible to use for such systems biopolymers as proteins, etc.?

We thank the referee for their suggestion. We have added a paragraph expanding the applicability of these systems in biopolymers in page 19.

2) I highly recommend summarizing various applications with an exiting examples as a Picture somewhere in the review.

The observation of the referee was considered by the authors on a first instance as well. However, when looking for illustrative pictures in each application field we found that, due to the scarce literature available for some applications (e.g. slippery surfaces or stretchable electronics), nice and clear visual examples were unfortunately not found. Thus, schematic representations concerning the different applications evaluated were included in the graphical abstract for this purpose.

3) It will be very good also summarize the synthesis methods of magnetic self-healing systems and widely used polymers and surfactants in one or several Tables. 

We thank the referee for their sharp comment. We have followed their suggestion and we have added two new tables summarizing on the one hand, the different self-healing mechanisms and magnetic fillers as a function of the type of the fabrication method in table 2, and on the other hand, an exemplification of the different self-healing mechanisms, magnetic fillers and fabrication methods reported for the evaluated magnetic self-healing application fields have been described in table 3.

Round 2

Reviewer 4 Report

Dear authors,

Thank you for the revised version of the paper. I have read your text with great pleasure. You have essential examples in the most sections. However, my first question about magnetic self-healing polymers was just the interest is there is any exiting examples of magnetic self-healing polymers for drug delivery. The presented in literature data is just smart magnetic systems. But I haven't seen that somebody has done something differ than simple magnetic nanoparticles. The example of magnetic nanoparticles which have extraordinary properties because of the self-healing and magnetism will be essential but optional.

I think the paper may be accepted in present form.